# Diagnosis of Simultaneous Atrial and Ventricular Mechanical Performance in Patients with Systemic Sclerosis

**DOI:** 10.3390/biology11020305

**Published:** 2022-02-14

**Authors:** Mohammadbagher Sharifkazemi, Mohammadali Nazarinia, Alireza Arjangzade, Mohamad Goldust, Zahra Hooshanginezhad

**Affiliations:** 1Department of Cardiology, Nemazee Hospital, Shiraz University of Medical Sciences, Shiraz 7193613311, Iran; sharifkm@sums.ac.ir (M.S.); arjangzade@sums.ac.ir (A.A.); 2Department of Internal Medicine, Shiraz University of Medical Sciences, Shiraz 7193613311, Iran; nazariniam@sums.ac.ir; 3Department of Dermatology, University Medical Center of the Johannes Gutenberg University, 55131 Mainz, Germany; mgoldust@uni-mainz.de

**Keywords:** sclerosis systemic, speckle tracking echocardiography, doppler color

## Abstract

**Simple Summary:**

Systemic Sclerosis (SSc) is a chronic connective tissue disorder with an inflammatory and autoimmune nature. The disease presents with microvascular changes, endothelial cell dysfunction, and fibrosis in visceral organs and tissues including lung, skin and heart. Cardiac involvement is a predictor of poor prognosis in this disease; therefore, early and pre-clinical diagnosis of cardiac involvement can be helpful in management of SSc. Two-dimensional Speckle Tracking Echocardiography (2D-STE) is a new method for the evaluation of myocardial strain in longitudinal, circumferential and radial planes. This study aimed at evaluating the mechanical performance of all cardiac chambers by using 2D-STE in patients with scleroderma, and its comparison with normal individuals. This study demonstrated that using 2D-STE can result in the diagnosis of impaired chamber mechanics and function in subclinical stages. Based on our findings, the simultaneous evaluation of all cardiac chambers by 2D-STE provides valuable information regarding myocardial involvement in patients with SSc.

**Abstract:**

**Objective**: Cardiac involvement is a predictor of poor prognosis in patients with systemic sclerosis (SSc); therefore, preclinical diagnosis of heart involvement is crucial. Two-dimensional speckle tracking echocardiography (2D-STE), a method for evaluating the myocardial strain, could be helpful for the early diagnosis of cardiac mechanical function abnormalities. In this study, the simultaneous evaluation of all cardiac chambers was studied in patients with SSc, compared with normal individuals. **Methods:** The results of transthoracic echocardiography (TTE) and 2D-STE of 37 patients with SSc and 37 healthy individuals and the longitudinal strain (LS) of all chambers was precisely evaluated. The collected data were analyzed using SPSS version 16, and independent-sample *t* test and Chi-square test were used for comparison between the groups. **Results:** The mean ± SD of the participants’ age was 45.7 ± 11.54 (range of 17 to 68) years; most of them were women (75.7%). TTE showed higher left atrial (LA) volume (*p* < 0.001), right atrial (RA) area (*p* = 0.007), the severity of ventricular dysfunction (*p* < 0.05) and inferior vena cava diameter (*p* = 0.005), compared with the control group. Spectral and tissue Doppler echocardiography showed higher systolic pulmonary arterial pressure (sPAP) and mitral A wave velocity, and lower E/A ratio, E’ velocity of left ventricular (LV) septal and lateral wall in the case group (*p* < 0.05). Color Doppler echocardiography showed a higher frequency of valvular regurgitation in the case group (*p* < 0.05). The results of 2D-STE showed lower LA roof LS (*p* < 0.001), LA average LS (*p* = 0.015), LA global LS (*p* = 0.028), and LA ejection fraction (LAEF) (*p* = 0.001), lower mean RA left wall (*p* = 0.048) and EF (*p* < 0.001), and higher RV global LS in the case group (*p* = 0.025). **Conclusions:** Simultaneous evaluation of all cardiac chambers by 2D-STE provides valuable information about the myocardial involvement in patients with SSc. Therefore, it is suggested to use this method for the early diagnosis of cardiac involvement in such patients.

## 1. Introduction

Systemic Sclerosis (SSc) is a complex chronic connective tissue disorder, presenting with microvascular changes, endothelial cell dysfunction, and fibrosis in visceral organs and tissues, as well as complex autoimmune responses [1]. The prevalence rates reported for the two major subtypes, SSc with limited or diffuse cutaneous involvement, vary according to geographical and racial risk factors, as well as the diagnostic criteria used in each study, while female predominance is a common statement [2]. Patients with SSc have a 2.7– to 3.4–fold increased mortality rate compared to the general population [3] with poorer prognosis in patients with diffuse cutaneous SSc (dcSSc), pulmonary, renal, and cardiac involvements [4]. The cardiac involvement of SSc that results in a five–fold increased mortality rate [4] and sudden cardiac death in 21%–54% of patients [5], can be caused primarily by the inflammatory induced changes in cardiac structures, such as microvasculature, pericardium, conduction system, and valves, resulting in arrhythmia, pericarditis, valvular diseases, cardiac failure and cardiomyopathy, and/or secondarily by interstitial lung disease, pulmonary arterial hypertension (PAH) and kidney involvement [6]. The diagnosis of cardiac involvement in SSc requires several medical specialties, including rheumatologists and cardiologists, and because of the non-specific nature of the clinical presentations, screening methods, such as electrocardiography (ECG), computed tomography, magnetic resonance imaging (MRI) and transthoracic echocardiography (TTE), are recommended for early diagnosis of cardiac involvement in patients with SSc, especially in patients with risk factors [7]. Moreover, because of the low costs, availability and ease of performance of echocardiography, it is recommended as the primary step for detection and also recommended is at least annual follow-up of myocardial, pericardial, and valvular involvements in the patients with SSc [8]. 

Two-dimensional speckle tracking echocardiography (2D-STE), which generates frame-by-frame patterns in different directions and amplitudes of motion and demonstrates myocardial deformation throughout the cardiac cycle [9], is suggested as an appropriate tool for the diagnosis of ventricular dysfunction, not detectable by conventional echocardiography [10,11,12]. Furthermore, 2D-STE can detect the early changes of atrial mechanical function, which can be an early sign of diastolic dysfunction (DD) and cardiac involvement in patients with SSc and is useful for risk stratification [13,14,15]. 2D-STE is already available in almost all well-equipped echocardiographic laboratories. It is a non-invasive method and it is not a time-consuming procedure. It takes less than one hour for an expert echocardiographer to evaluate meticulously the mechanical performance of all cardiac chambers [14,15].

Despite the systemic nature of SSc that can involve any cardiac chamber, the previous studies were limited to the evaluation of one chamber (atrium or ventricle) or one side (left or right). Therefore, in the present study, we aimed to perform a detailed four-chamber 2D-STE for the accurate evaluation of four-chamber function in patients with SSc, compared with the normal population. The study was confirmed by the ethics committee of Shiraz University of Medical Science.

## 2. Methods

In the present study, patients who were diagnosed with Scleroderma by a rheumatologist (according to 2013 ACR/EULAR SSc Classification Criteria [16]) and referred to us by Hafez Hospital, Shiraz, Iran, from November 2019 to May 2020 were considered as the case group, and the control group was conveniently selected from healthy volunteers, who were referred to the Shahid Faghihi Hospital echocardiography laboratory. Any individual with signs of severe heart failure, interstitial lung disease, severe pulmonary hypertension and lung involvement, a positive history of congenital heart disease, chronic obstructive pulmonary disease, pulmonary thromboembolism, cardiac surgery, percutaneous coronary intervention (PCI), and using a pacemaker were excluded from this study. The sample size was calculated at 33 in each group with a ratio of 1:1, considering error at 5%, power at 80%, and an effect size of about 60% or a risk ratio of about 3 in the two groups. Considering the chance of being lost to follow–up, 40 individuals were recruited in each group. Eligible individuals were enrolled into the study by the census method after they were informed about the study objectives and completed the written informed consent form.

The participants’ demographic characteristics, including sex, age, weight, height and length of disease were recorded, and their body mass index (BMI) and body surface area (BSA) were calculated using standard formulas. Their history of chronic diseases, such as hypertension (with maximum mild left ventricular hypertrophy [LVH] in TTE), hyperlipidemia, hypothyroidism, ischemic heart disease (with no history of previous PCI or coronary artery bypass graft [CABG]), and diabetes mellitus were recorded. It should be noted that we tried to choose our control group with the same medical histories as the case group.

All individuals underwent TTE and 2D-STE using a Siemens Acuson sc2000 PRIME ultrasound machine at Shahid Faghihi Hospital Echocardiography Center by one cardiologist. We utilized the eSie Velocity Vector Imaging (VVI) application, designed for quantitative and visual assessment of cardiac dynamics on Acuson sc2000 PRIME for STE. All participants were examined in the left lateral decubitus position using the grey-scale 2D imaging technique with adjustment of image contrast, frequency, depth, and sector size for adequate frame rate and optimal images. All images and measurements were obtained from the standard views with stable ECG. Offline analysis of 2D images was performed precisely in an echocardiography workstation for measuring the following parameters in proper time: interventricular septum (IVS) thickness, ventricular and atrial diameters, aorta diameter at sinuses, left ventricular (LV) ejection fraction (EF), and RV fractional area change (FAC).

Tricuspid annular plane systolic excursion (TAPSE) was measured through M–mode images. Left atrial (LA) volume and right atrial (RA) area were measured with the apical monoplane method in four-chamber view at end-systole (maximal LA and RA size). The function of each valve was evaluated by the color Doppler method in different views and was reported according to the 2020 ACC/AHA guideline for the management of patients with valvular heart disease. The mitral, aortic, and tricuspid valve regurgitation were assessed and categorized as “at risk”, “progressive”, “asymptomatic severe” and “symptomatic severe” [17]. Inferior vena cava (IVC) diameter measured in long axis subcostal view at 1 to 2 cm from its junction to the right atrium. The systolic pulmonary artery pressure (sPAP) was estimated using modified Bernoulli’s formula from peak tricuspid regurgitation jet velocities plus adding right atrial pressure. Right atrial pressure was estimated from IVC diameter and respiratory collapsibility (IVC diameter < 2.1 with more than 50% collapse with a sniff suggests normal RA pressure of 3 mmHg, and IVC diameter > 2.1 with less than 50% collapse while sniffing suggests high RA pressure of 15 mmHg [18]). 

The spectral Doppler parameters were evaluated including mitral valve (MV) E and A points velocity and E/A ratio. The LV diastolic function was determined from the mitral flow velocity pattern by pulsed Doppler echocardiography, in addition to the tissue, Doppler parameters measured included LV lateral e’ and LV septal e’. 

To assess the myocardial function of atria and ventricles, the endocardial border was manually traced using a point and click technique. The eSie VVI application automatically generated a wide range of interest and was re-adjusted manually, which included dividing the atria into three segments: (1) left free wall, (2) right free wall, and (3) roof; generating the average and global longitudinal strains, summarizing all the data in a table per atrium. 

To evaluate the ventricular strain, we divided each ventricular chamber into six regions: (1) lateral base region, (2) lateral mid-region, (3) lateral apex region, (4) septal apex region, (5) septal mid-region, and (6) septal base region. After calculating each region’s strain, the average and global strain were also calculated for each ventricular chamber.

The patients with moderate-to-severe left ventricular hypertension, pericardial effusion, or poor quality of echocardiographic images for the evaluation of the endocardial border because of the patient’s poor echo window or tachypnea were excluded from the study. A total of 81 volunteers met the eligibility criteria for the study, 40 patients with SSc and 41 healthy volunteers, and were introduced to the echocardiography laboratory. Seventy-four volunteers were included in this study, as seven cases had poor image qualities and were excluded (Figure 1).

## 3. Statistical Analysis

The descriptive statistics included mean and standard deviation (SD) for the quantitative variables and number (percentage) for the qualitative variables. The normality of the distributions was evaluated using frequency histograms and the Shapiro–Wilks test. The comparison between the groups was performed using the independent samples *t* test and Chi-square test for the quantitative and qualitative variables, respectively. For data analysis, SPSS for Windows, Version 16.0. (SPSS Inc. Released 2007. Chicago, IL, USA) was used. The *p* value < 0.05 was considered statistically significant. 

## 4. Results

A total of 74 participants completed the study, 37 in each group. The mean ± SD of the participants’ age was 45.7 ± 11.54 (range of 17 to 68) years; most of our participants were women (75.7%). As shown in Table 1, the participants of the two groups had no significant differences in terms of mean age, mean BMI, sex distribution, and frequency of underlying diseases (*p* > 0.05; Table 1), while BSA was different between the two study groups (*p* = 0.02; Table 1). The mean ±SD of the disease duration was 9.62 ± 6.02 months, and the lung involvement was seen in 59.5% of patients with SSc (*n* = 22). Among the 37 Scleroderma patients who enrolled in our study, 23 (62%) were on calcium channel blockers (CCB), 33 (89%) were on Immunosuppressive therapy, nine (24%) were using endothelin receptor antagonists and four (10%) used Angiotensin II Receptor Blockers.

According to the results of TTE, as shown in Table 2, no statistically significant difference was observed between the case and control groups in LA systolic diameter in parasternal long-axis view, aorta size, IVS thickness, LV diameter, LVEF, FAC, and TAPSE; however, the case group had a significantly higher LA volume (*p* < 0.001), RA area (*p* = 0.007) and IVC diameter (*p* = 0.005) compared with the control group.

According to the results of spectral and tissue Doppler echocardiography, the case group had a statistically higher estimated sPAP and mitral A wave velocity (*p* < 0.05; Table 3) and lower LV inflow E/A ratio, e’ velocity of LV septal and lateral wall, compared with the control group (*p* < 0.05; Table 3). Considering PAH, mild, moderate and severe PAH were observed in 32%, 14%, and 6% of the case group, while none in the control group had PAH (*p* = 0.001).

According to color Doppler echocardiography results, 16.2% of the case group had progressive (stage B) and 8.1% had severe (stages C, D) tricuspid regurgitation, respectively, while only stage A (at risk) of tricuspid regurgitation was observed in the control group (37.8%; *p* < 0.001). The frequency of stage A and B mitral valve regurgitation was also higher in the case group compared with the control group (37.8% vs. 8.1%; *p* = 0.002), and there were no cases of severe mitral valve regurgitation in our study population. The aortic regurgitation was only observed in two patients (5.4%) of the case group, and there were no cases of progressive or severe aortic valve regurgitation in our study population (as mentioned before the severity stages defined according to the 2020 ACC/AHA guideline for the management of patients with valvular heart disease).

The strain results of STE showed lower LA roof longitudinal strain (*p* < 0.001), LA average longitudinal strain (*p* = 0.015), LA global longitudinal strain (*p* = 0.028), and LAEF (*p* = 0.001) in the case group (Table 4). Considering RA parameters, there was a statistically significant discrepancy in RA left wall (*p* = 0.048) and RAEF (*p* < 0.001) between the two groups (Table 4). RV global strain was also higher in the case group than that of the control group (*p* = 0.025; Table 4). However, there was no statistically significant difference between the two groups in LV parameters (*p* > 0.05; Table 4). 

## 5. Discussion

Simultaneous four-chamber evaluation of patients with SSc, compared with a healthy control group, using TTE and 2D-STE in the present study, had two important findings. The first important finding of the present study was the detection of cardiac changes in patients with SSc by 2D-STE, which were undetectable by TTE; the second one is related to the simultaneous evaluation of the four cardiac chambers, considering the principal role of atria in cardiac forward stroke volume [19]. Although alterations of LV and RV structure and function are a well-known issue in patients with SSc and the focus of attention in previous studies [10,11,12,20], the simultaneous evaluation of both atrial structure and function in this systemic disease, SSc, has not been evaluated previously, as far as the authors are concerned. 

In the present study, TTE showed greater LA volume, RA area, and IVC diameter; moreover, Doppler results showed a higher frequency and severity of LV diastolic dysfunction (DD) and PAH in patients with SSc. However, other TTE parameters were not different between the groups. The results of STE also showed that the mechanical function of LA (LA EF) and longitudinal strain in LA roof, global, and average longitudinal strain were markedly reduced, even in patients with preserved LV systolic function. Furthermore, the STE parameter indicates the involvement of LA with SSc, independent of the LV systolic function. They can also develop these mentioned right atrial changes secondary to elevated sPAP [21], notably right ventricular hypertrophy, because PAH re-emphasizes the crucial role of the right atrial booster pump, in order to maintain the adequate ventricular filling [22]. Therefore, the atrial systolic or diastolic dysfunction not only dictates a higher mean atrial pressure, but also predisposes the patients to pulmonary vein or systemic vein hypertension [23]. The impaired atria in these patients are supposed to develop because of the fibrosis and inflammatory changes, resulting in negative remodeling in atrial myocardium [24]. Interestingly, in our study, results of STE showed reduced mechanical function (RA EF) in patients with SSc, while no statistically significant difference was detected in RA global and longitudinal strain between patients with SSc and the control group, except in the RA roof wall. This is while others have confirmed significant impairment of RA function and enlargement, determined by 2D-STE in patients with SSc, especially those with pulmonary fibrosis and increased sPAP [13,15]. D’Andrea and colleagues referred to the impairment of RA function and enlargement, detected by STE, even in patients without PAH. They also suggested that RA strain was associated with several factors, such as RA area, LV stroke volume and IVC diameter [25]; therefore, the variability of RA strain predictors can justify our findings, despite the small study population.

Previous studies have also confirmed impaired LA function in patients with SSc when evaluated by 2D-STE, which emphasizes the role of this imaging modality in the diagnosis of LA dysfunction in such patients [14,15]. In the study by Ataş and his colleagues (Marmara University; Kadıköy/İstanbul, Turkey), the results showed impaired LA pump function and lower strain in patients with SSc, which could not be justified by the LVDD in this group, while LV systolic function was also not different between the patients with SSc and the control group [14]. These results confirmed the atrial changes induced by SSc, shown in the results of the present study. Agoston and colleagues also included patients with SSc with no clinical signs of cardiac involvement and reported that the mean LVEF of these patients was within a normal range (63.1 ± 4%) and TTE showed no LA dysfunction, while LA strain values indicated impaired reservoir and conduit function of LA in such patients [26]. No difference in LVEF, RV FAC, and TAPSE in the TTE results of the present study between the case and control groups could also be because of the fact that we excluded patients with signs of severe heart failure from our study. It should also be noted that, because the study period overlapped with the Coronavirus disease 2019 (COVID–19) pandemic, several patients, including high-risk patients and those with severe SSc, refused to participate in this study.

Tissue and spectral Doppler results in the current study revealed higher mitral A wave velocity and lower E/A ratio, e’ velocity of LV septal and lateral wall in patients with SSc, which indicated a higher prevalence of LVDD in patients with SSc, compared with the control group. Studying 153 consecutive patients with SSc showed LVDD in 23% of patients, while LV systolic dysfunction was only present in 5.2% of patients with SSc [27], which is consistent with the results of the present study. Keep in mind that LVDD is suggested to be a significant predictor of mortality in patients with SSc and is thus of great importance [28]. Decreased e’ velocity has been associated with SSc disease duration, age, history of coronary artery disease and systemic hypertension, and each standard deviation (SD) decrease in e’ velocity results in a 3.2–fold increased mortality in such patients [27]. It is not only the above-mentioned (primary and secondary) pathologies that induce myocarditis, some SSc treatments can also have cardiotoxic effects and result in ventricular systolic and/or diastolic dysfunction [29]. Therefore, it is suggested to consider the risk factors of cardiac involvement in patients with SSc and perform periodic follow-up for cardiac assessment [8]. The use of 2D-STE can be an appropriate tool for the early diagnosis of LV dysfunction and follow-up of patients, not only for the treatment of the cardiac involvement but also for the adjustment of anti-SSc treatments [30].

The biventricular analysis in the current study has yielded that RV global longitudinal strain has declined, whereas the LV global longitudinal strain has been preserved. Previously, RV impairments were thought to occur as a result of PAH and pressure overload [10,31], and the results of the present study also showed different sPAP values between the two study groups. However, considering biventricular changes, studies evaluating the results of biventricular STE have shown diverse results, which could be because of the different inclusion criteria considered for patients’ enrollment. Karadag and colleagues reported lower LV global longitudinal strain (GLS) in patients with SSc without overt cardiac disease, compared with the control group, while they reported no difference in RV GLS between the groups [31]. These results are exactly the opposite of our results; however, they claimed that several factors attributed to ventricular strain, such as C–reactive protein [32], the difference in which among the study population can be the source of different results. Interestingly, Guerra and others showed biventricular impairment in patients with SSc without overt clinical involvement, considering GLS measurement, with a homogeneous myocardial deformation pattern in RV and an eccentric one in LV [12]. Considering the discrepancy of the results of the previous studies, more studies are required in this regard. This is while most studies have only evaluated the LV in patients with SSc [30,33].

The present study was limited in terms of sample size, sex disproportion of the study population, non-randomized inclusion of patients in the study and the two groups, the exclusion of patients with overt heart failure (which could have affected our strain measurements), and the evaluation of PAH and diastolic function using non-invasive methods (Doppler Echocardiography) rather than the gold standard methods of evaluation, that is, catheterization. The lack of long-term follow-up was another limitation in the present study, and we could not identify whether the atrial impairments were associated with future ventricular dysfunction or not.

## 6. Conclusions

The results of the present study showed that the simultaneous evaluation of the four chambers using 2D-STE resulted in the diagnosis of impaired either mechanics or function of the chambers in subclinical stages and even more apparently, the changes in the left atrium before any involvement of the left ventricle in TTE or STE. Further studies are required to confirm these results and evaluate the association of these impairments with patients’ morbidity and mortality rates.

## Figures and Tables

**Figure 1 biology-11-00305-f001:**
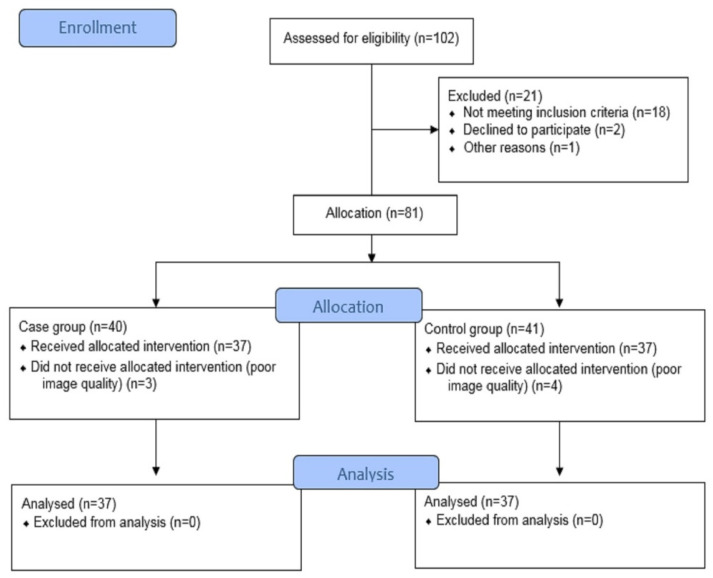
Enrollment status.

**Table 1 biology-11-00305-t001:** Comparing the demographic characteristics of the two study groups.

Variables	Categories	Case Group (*n* = 37)	Control Group (*n* = 37)	Statistics	*p* Value
**Age (years), mean ± SD**		46.97 ± 11.15	44.43 ± 11.93	0.946	0.347 ^†^
**Sex, *n*.(%)**	**Female**	26 (70%)	24 (64.9%)	0.958	0.421 *
**Duration of the disease (months)**	**Male**	11 (29%) 9.62 ±6.02	13 (35.1)	0.958	
**Body mass index (kg/m^2^), mean ± SD**		23.98 ± 3.84	25.45 ± 3.32	1.75	0.083 *
**Body surface area (m^2^), mean ± SD**		1.66 ± 0.15	1.76± 0.13	3.13	0.002 *
**Hypertension, N.(%)**		8 (21.6%)	5 (13.5%)	0.84	0.359 ^†^
**Hyperlipidemia, N.(%)**		1 (2.7%)	0 (0%)	1.01	0.314 ^†^
**Hypothyroidism, N.(%)**		1 (2.7%)	0 (0%)	0	>0.999 ^†^
**Ischemic heart disease, N.(%)**		1 (2.7%)	1 (2.7%)	0	>0.999 ^†^
**Diabetes mellitus, N.(%)**		1 (2.7%)	0 (0%)	1.01	0.314 ^†^
**Consumption of BBs**		0	0		
**Consumption of CCBs**		23 (62%)	0		
**Consumption of Immuno supressants**		33 (89%)	0		
**Consumption of Endothelin Receptor Antagonists**		9 (24%)	0		
**Consumption of Angiotensin II Receptor Blockers**		4 (10%)	0		

* The results of student *t* test; ^†^ The results of Chi-square test, BB: Beta blocker, CCB: Calcium channel blockers.

**Table 2 biology-11-00305-t002:** Comparing the results of two-dimensional echocardiography parameters between the two study groups.

Variables	Case Group (*n* = 37)	Control Group (*n* = 37)	Statistics	*p* Value *
**Left atrial systolic diameter (mm)**	29.72 ± 3.72	28.97 ± 3.04	0.957	0.342
**Left atrial volume (cm^3^)**	40.02 ± 9.51	30.02 ± 7.28	5.07	<0.001
**Left atrial volume (cm3) indexed with BSA**	24.32 ± 6.14	16.75 ± 4.99		<0.001
**Aorta at sinuses diameter (mm)**	26.59 ± 3.10	25.88 ± 1.86	1.18	0.243
**Interventricular septum thickness (mm)**	9.02 ± 1.3	9.08 ± 1.06	0.196	0.845
**Left ventricular internal dimension, systolic (mm)**	27.64 ± 5.73	28.7 ± 1.98	1.05	0.296
**Left ventricular internal dimension, diastolic (mm)**	45.54 ± 5.07	46.27 ± 1.92	0.81	0.416
**Left ventricular ejection fraction (%)**	54.91 ± 4.57	55.97 ± 1.18	1.18	0.240
**Right atrial area (cm^2^)**	14.32 ± 6.03	11.43 ± 2.07	2.75	0.007
**Right atrial area (cm^2^) indexed with BSA**	8.67 ± 3.70	6.48 ± 1.22		0.001
**Right ventricular fractional area change (%)**	38.44 ± 8.03	41.18 ± 4.85	0.525	0.092
**Tricuspid annular plane systolic excursion (cm)**	19.37 ± 2.88	19.97 ± 1.23	3.69	0.252
**Inferior vena cava diameter (mm)**	16.18 ± 1.46	15.18 ± 1.52	2.87	0.005

* The results of student *t* test; all values are reported as mean ±SD.

**Table 3 biology-11-00305-t003:** Comparing the results of two-dimensional echocardiography spectral and tissue Doppler parameters between the two study groups.

Spectral and tissue Doppler Parameters	Case Group (*n* = 37)	Control Group (*n* = 37)	Statistics	*p* Value *
**Estimated systolic pulmonary artery pressure (mmHg)**	31.67 ± 15.02	20.82 ± 2.5	4.3	0.001
**Mitral valve E point velocity (cm/s)**	71.51 ± 18.26	67.72 ± 12.26	1.06	0.289
**Mitral valve A point velocity (cm/s)**	75.89 ± 17.61	61.08 ± 9.24	4.52	<0.001
**Mitral valve E/A**	0.95 ± 0.34	1.130 ± 0.28	2.36	0.021
**Left ventricular septal e’ velocity (cm/s)**	7.75 ± 2.26	9.83 ± 1.81	4.35	<0.001
**Left ventricular lateral e’ velocity (cm/s)**	11.83 ± 3.09	14.78 ± 2.79	4.3	<0.001

* The results of student *t* test; all values are reported as mean ±SD.

**Table 4 biology-11-00305-t004:** Comparing the longitudinal strain parameters in speckle tracking echocardiography between the two study groups.

	Variables	Case Group (*n* = 37)	Control Group (*n* = 37)	Statistics	*p* Value *
**Left trial**	Left wall	36.99 ± 17.45	41.74 ± 21.56	1.02	0.309
	Left roof	42.48 ± 26.26	64.49 ± 29.85	4.06	<0.001
	Right wall	31.93 ± 17.82	28.74 ± 21.96	0.675	0.502
	Average	36.96 ± 14.87	46.73 ± 18.33	2.49	0.015
	Global	35.97 ± 14.80	45.01 ± 19.33	2.33	0.028
	Ejection fraction	62.19 ± 13.01	71.21 ± 8.18	3.55	0.001
**Right atrial**	Left wall	33.39 ± 21.57	41.82 ± 14.56	2.08	0.048
	Left roof	45.76 ± 30.19	42.47 ± 22.62	0.539	0.591
	Right wall	83.66 ± 59.19	88.91 ± 40.52	0.454	0.651
	Average	54.34 ± 28.08	57.67 ± 18.18	0.795	0.429
	Global	51.93 ± 27.97	55.88 ± 18.88	0.718	0.475
	Ejection fraction	58.25 ± 14.13	68.64 ± 8.05	3.92	<0.001
**Left ventricle**	Average strain	−18.34 ± 4.35	−18.06 ± 3.49	0.299	0.796
	Global strain	−17.85 ± 4.07	−17.83 ± 2.93	0.015	0.988
	Ejection fraction	54.91 ± 4.57	55.97 ± 1.18	1.18	0.240
**Right ventricle**	Average strain	−18.44 ± 7.17	−17.75 ± 11.83	0.302	0.763
	Global strain	−16.93 ± 6.42	−19.57 ± 5.18	2.23	0.025
	Ejection fraction	38.44 ± 8.03	41.18 ± 4.85	1.68	0.092

* The results of student *t* test; all values are reported as mean ±SD.

## Data Availability

The data presented in this study are available on request from the corresponding author. The data are not publicly available due to our patients’ privacy.

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
