# Peer review of "Diagnosis of Simultaneous Atrial and Ventricular Mechanical Performance in Patients with Systemic Sclerosis"

_biology, 2022, doi:10.3390/biology11020305_

Round 1

Reviewer 1 Report

This study is about the simultaneous evaluation of the four chambers of the heart using 2D-STE in patients with systemic sclerosiss.  The topic is relevant to clinical practice.

The English language usage is good.

The statistical method is clean and straightforward

Although the sample size is relatively small, the distribution is adequate for the disease demographics studied.

I have some minor notes.

Data might be added if the 2D STE is available in all cardiac echocardiography machines and centers?

How much more time does it take to examine all four chambers as described in the Methods?

The title is very long.

This article might be added to the references:

Porpáczy A, Nógrádi Á, Kehl D, Strenner M, Minier T, Czirják L, Komócsi A, Faludi R. Impairment of Left Atrial Mechanics Is an Early Sign of Myocardial Involvement in Systemic Sclerosis. J Card Fail. 2018 Apr;24(4):234-242. doi: 10.1016/j.cardfail.2018.02.012. PMID: 29501919. 

Author Response

Dear reviewer,

Thank you for the comments on our manuscript. We have addressed the your comments in the manuscript and replied each below.

Reviewer Comments:

This study is about the simultaneous evaluation of the four chambers of the heart using 2D-STE in patients with systemic sclerosiss.  The topic is relevant to clinical practice.

The English language usage is good.

The statistical method is clean and straightforward

Although the sample size is relatively small, the distribution is adequate for the disease demographics studied

I have some minor notes.

Data might be added if the 2D STE is available in all cardiac echocardiography machines and centers?

2D STE is already available in almost all well-equipped echocardiographic laboratories. In the last edition of American Society of Echocardography textbook(chapter 25, the third edition; 2022) the writer concluded that because of numerous weakpoints regarding ejection fraction, STE can be used as an effective supplemental method for better estimation of chambers function.

How much more time does it take to examine all four chambers as described in the Methods?

 It is not a time-consuming procedure. In other words, it takes less than one hour to evaluate meticulous mechanical performance of all cardiac chambers by an expert echocardiographer

The title is very long.

Diagnosis of Simultaneous Atrial and Ventricular Mechanical Performance in Patients with Systemic Sclerosis

This article might be added to the references:

Porpáczy A, Nógrádi Á, Kehl D, Strenner M, Minier T, Czirják L, Komócsi A, Faludi R.. J Card Fail. 2018 Apr;24(4):234-242. doi: 10.1016/j.cardfail.2018.02.012. PMID: 29501919. 

We have reviewed Dr Porpáczy and his colleagues study on LA strain and its ability to predict myocardial involvement in early stages of scleroderma patients. Therefore we had already cited their precious study in the manuscript(reference NO.15)

Reviewer 2 Report

The author's paper is very interesting, but I have a few questions and things I would like to see clarified.

Please confirm with your rheumatologist which diagnostic criteria you used to diagnose SSc and state which criteria you used in your paper.

In Table 1, please add information about the patient's medications (e.g., whether beta blockers or endothelin receptor antagonists are used).

Author Response

Dear reviewer,

Thank you for the comments on our manuscript. We have addressed the your comments in the manuscript and replied each below.

Reviewer Comments:

The author's paper is very interesting, but I have a few questions and things I would like to see clarified.

Please confirm with your rheumatologist which diagnostic criteria you used to diagnose SSc and state which criteria you used in your paper.

We enrolled patients who diagnosed with Scleroderma according to the clinical features of the disease, associated with positive SSc-associated autoantibodies test and nailfold capillaroscopy and they classified according to the 2013 American College of Rheumatolgy (ACR)/European League Against Rheumatism (EULAR) guideline.

In Table 1, please add information about the patient's medications (e.g., whether beta blockers or endothelin receptor antagonists are used).

Among the 37 Scleroderma patients who enrolled our study 23 (62%) was on Calcium Channel Blocekrs, 33 (89%) was on Immunosuppressive therapy, 9 (24%) was on using Endothelin Receptor Antagonists and 4 (10%) has used Angiotensin II Receptor Blockers. (Answer to the reviewer’s comment have been reflected in the Table 1 in the latest version.)